# Less is More: Feature Selection for Adversarial Robustness with Compressive Counter-Adversarial Attacks

**Emre Ozfatura** [1]  **Muhammad Zaid Hameed** [2]  **Kerem Ozfatura** [1]  **Deniz Gunduz** [1]

## Abstract

A common observation regarding adversarial attacks is that they mostly give rise to false activation at the penultimate layer to fool the classifier. Assuming that these activation values correspond to certain features of the input, the objective becomes choosing the features that are most useful for classification. Hence, we propose a novel approach to identify the important features by employing counter-adversarial attacks, which highlights the consistency at the penultimate layer with respect to perturbations on input samples. First, we empirically show that there exist a subset of features, classification based in which bridge the gap between the clean and robust accuracy. Second, we propose a simple yet efficient mechanism to identify those features by searching the neighborhood of input sample. We then select features by observing the consistency of the activation values at the penultimate layer.

## 1. Introduction

Despite their remarkable performance in a wide range of real world problems, deep neural networks (DNNs) have been shown to be vulnerable to adversarial attacks, where a small perturbation to the input data can fool the network (Bruna et al., 2014; Goodfellow et al., 2015; Carlini & Wagner, 2017). Consequently, there has been a lot of work in building robust models against these adversarial examples (Madry et al., 2018; Zhang et al., 2019; Shafahi et al., 2019; Qin et al., 2019; Sehwag et al., 2020; Wu et al., 2020; Gowal et al., 2020). The most successful approach to building robust DNN models is based on adversarial training (AT) (Madry et al., 2018), where a network is trained on (approximate) worst-case adversarial examples (often generated by iteratively maximizing some loss function). Despite the

success of AT in increasing the robustness, the performance gap between the robust accuracy (adversarial examples as input) and the clean accuracy (non-adversarial inputs) of an adversarially trained model is still quite large (Madry et al., 2018; Zhang et al., 2019; Tsipras et al., 2019; Yang et al., 2020). Several modifications of AT have been proposed to improve robustness, and to bridge the gap between the robust and clean accuracies e.g., by label smoothing and stochastic weight averaging (Chen et al., 2021), by using additional unlabelled data (Carmon et al., 2019), and by employing different activation functions (Xie et al., 2020). However, these techniques result in marginal increases in robust accuracy, and bridging the gap between robust and clean accuracies has so far remained elusive.

In a parallel line of research, it has been argued that adversarial examples are actually characteristics of the datasets used for training (Ilyas et al., 2019). That is, the datasets contain samples with both robust and non-robust features, and adversarial examples exist due to the presence of these non-robust features, which are exploited by the classifier to improve its accuracy when trained on the dataset. It has been further shown that an adversarially trained network (indirectly) limits the effect of these non-robust features. Based on this observation, minimizing the influence of non-robust features could pave the way to building a robust model, but until recently very little work is done in this area (Bai et al., 2021; Yan et al., 2021; Xie et al., 2019). Note that, feature selection is not a new problem in machine learning (Chen et al., 2018a; Gao et al., 2016; Shrikumar et al., 2017; Brown et al., 2012); however, existing schemes are generally used for model interpretability and have not been investigated for robustness against adversarial attacks.

We argue that the key limitation of AT as a defence mechanism is being oblivious to intrinsic properties of the observed sample. By intrinsic properties we refer to the impact of perturbations in the image domain to the distribution of the activation values at the penultimate layer. Assuming that each activation value corresponds to a different feature, the change in the distribution of the activation values due to variations within a close neighbourhood of the image provides an insight on their consistency, and helps to identify the common features in the neighbourhood. Hence, we argue that enforcing the classifier to perform predictions using only these common features will enhance its robustness.

To achieve this, we propose a latent masking approach for

[1]Information Processing and Communications Lab, Department of Electrical and Electronic Engineering, Imperial College London, UK [2]Resilient Information Systems Security Group, Department of Computing, Imperial College London, UK. Correspondence to: Emre Ozfatura <m.ozfatura@imperial.ac.uk>.

*Accepted by the ICML 2021 workshop on A Blessing in Disguise: The Prospects and Perils of Adversarial Machine Learning.*

feature selection that conditions the classifier, such that, based on the additional side information obtained by searching the close neighborhood of an input sample, the classifier utilizes only a subset of the activation values at the penultimate layer. The side information acts as a certain consistency measure on the activation values. We remark here that the majority of existing works approach the feature selection and activation masking problem by either identifying the robust features in the image domain or by analyzing the importance of each feature for the prediction of each class. Instead, our aim is to search for a robust representation at the penultimate layer for each image, which can be considered as a consensus representation for all the images within the close neighbourhood of the original image.

## 2. Preliminaries

### 2.1. Adversarial Training (AT)

Let $\mathbf{x} \in \mathbb{R}^n$ denote the input data that we want to classify to a set of labels $\mathcal{Y} = \{1, 2, \ldots, Y\}$. We define the classifier as a score function $F_{\boldsymbol{\theta}} : \mathbb{R}^n \times \mathcal{Y} \to \mathbb{R}$, which assigns label $\hat{y} \in \arg\max_{y \in \mathcal{Y}} F_{\boldsymbol{\theta}}(\mathbf{x}, y)$ to $\mathbf{x}$, where $\boldsymbol{\theta} \in \Theta$ denotes the parameters of this score function. With a slight abuse of notation, we also use $F$ to denote the classifier, $F(\mathbf{x})$ to denote the label assigned to $\mathbf{x}$, and $F(\mathbf{x}, y)$ to denote the score of class $y$ for input $\mathbf{x}$. For a DNN architecture, $F$ consists of $L$ layers, i.e., $F = f_L \circ f_{L-1} \circ \ldots \circ f_1$, where the last layer $f_L$ is a fully connected layer.

Let $\mathbf{x}$ be a correctly classified input, i.e., $y = F(\mathbf{x})$ is the true label. Furthermore, let $\mathcal{B}_\epsilon(\mathbf{x})$ denote the $L_p$-norm ball of radius $\epsilon$ centered at $\mathbf{x}$, i.e., $\mathcal{B}_\epsilon(\mathbf{x}) = \{\tilde{\mathbf{x}} : \|\tilde{\mathbf{x}} - \mathbf{x}\|_p \le \epsilon\}$. An adversarial attack on this classifier $F$ aims to modify the input $\mathbf{x}$ to $\tilde{\mathbf{x}} \in \mathcal{B}_\epsilon(\mathbf{x})$, such that $F(\tilde{\mathbf{x}}) \neq F(\mathbf{x})$. In practice, these adversarial examples are generated by optimizing some loss function $\mathcal{L}$ on the classifier (Carlini & Wagner, 2017; Moosavi-Dezfooli et al., 2016; Chen et al., 2018b; Laidlaw & Feizi, 2019; Madry et al., 2018).

AT has become *de facto* defense strategy against adversarial attacks, which can be considered as a two-player game formulated as the following min-max optimization problem:

$$\min_{\boldsymbol{\theta}} \max_{\tilde{\mathbf{x}} \in \mathcal{B}_\epsilon(\mathbf{x})} \mathcal{L}(F(\tilde{\mathbf{x}}, y), y), \qquad (1)$$

where $\tilde{\mathbf{x}}$ is the adversarial sample. Hence, during training, for each clean sample $\mathbf{x}$, first an adversarial sample that maximizes the loss $\mathcal{L}$ is chosen from the ball $\mathcal{B}_\epsilon(\mathbf{x})$, then the network is trained based on the adversarial sample for robustness at inference time.

### 2.2. Adversarial Examples and Activation Values

The impact of adversarial examples on the activation values in a DNN is previously studied in (Bai et al., 2021; Yan et al., 2021; Xiao et al., 2020). Let $\mathbf{z}^{(\mathbf{x})}$ denote the activation values at the penultimate layer for the input sample $\mathbf{x}$, i.e.,

$$\mathbf{z}^{(\mathbf{x})} = F_{\{L-1, \ldots, 1\}}(\mathbf{x}) = f_{L-1} \circ \ldots \circ f_1(\mathbf{x}). \qquad (2)$$

As highlighted in (Bai et al., 2021; Yan et al., 2021), when an adversarial example $\tilde{\mathbf{x}}$ is fed to network, one can observe a significant change in the distribution of the activation values in the penultimate layer compared to $\mathbf{x}$. By detecting and regulating these variations in the distribution of the activation values the robust accuracy can be improved (Bai et al., 2021; Yan et al., 2021).

We want to emphasize that previous works approached this problem by analyzing *the class-wise importance*, while we consider *sample-wise consistency*. To be more precise, previous works try to design an importance mask $\mathbf{m}$ for $\mathbf{z}^{(\tilde{\mathbf{x}})}$, such that $\mathbf{m}_i$ denotes the importance of the activation value[1] $\mathbf{z}_i^{(\tilde{\mathbf{x}})}$ for the prediction of a certain class. On the other hand, our objective is to measure the consistency of the activation values and design the mask accordingly. To clarify, let $\sigma_i$ be defined as $\sigma_i := |\mathbf{z}_i^{(\tilde{\mathbf{x}})} - \mathbf{z}_i^{(\mathbf{x})}|$, where $\tilde{\mathbf{x}} \in \mathcal{B}_\epsilon(\mathbf{x})$, then by sample-wise consistency we refer to a strategy where mask $\mathbf{m}$ is designed according to $\sigma_i$ values, particularly by choosing $\mathbf{m}_i$ inversely proportional to $\sigma_i$ for each image separately, independent from its class prediction. In the next section, we empirically show how such consistency measure could help to match the clean and robust accuracies in AT.

## 3. Matching Clean and Robust Accuracies

Assume that there exists an oracle $o$, which can access the clean sample $\mathbf{x}$ and the model $\boldsymbol{\theta}$, and hence, can obtain $\boldsymbol{\sigma}$ for a sample $\tilde{\mathbf{x}} \in \mathcal{B}_\epsilon(\mathbf{x})$ i.e., $\boldsymbol{\sigma} = o(\tilde{\mathbf{x}}; \mathbf{x}, \boldsymbol{\theta})$. Further, assume that the oracle shares only a partial side information $\mathbf{m}(\tilde{\mathbf{x}})$, where

$$\mathbf{m}(\tilde{\mathbf{x}}) = S_{topk}(\boldsymbol{\sigma}). \qquad (3)$$

Here, $S_{topk}(\mathbf{u})$ maps $\mathbf{u} \in \mathbb{R}^d$ to $\mathbf{m} \in \{0, 1\}^d$ such that $|\mathbf{m}|_0 = k$ and $\mathbf{m}_i = 1$ if $\mathbf{u}_i$ is one of the k-largest values in $\mathbf{u}$. Now, we argue that even side information $\mathbf{m}(\tilde{\mathbf{x}})$ might be sufficient to bridge the gap between the clean and robust accuracies. To verify this, we consider image classification on CIFAR-10 dataset (Krizhevsky, 2009) with the ResNet-18 model (He et al., 2016) and use side information $\mathbf{1}_d - \mathbf{m}(\tilde{\mathbf{x}})$, where $\mathbf{1}_d$ is a $d$-dimensional vector of ones, directly as a mask for the activation values in the penultimate layer $\mathbf{z}^{(\tilde{\mathbf{x}})}$. For AT, we follow (Madry et al., 2018) and use projected gradient descent (PGD) attack for 10 steps denoted by $\mathrm{PGD}_{10}$; see Appendix A for full details. Since classification is performed based on $(\mathbf{1}_d - \mathbf{m}(\tilde{\mathbf{x}})) \otimes \mathbf{z}^{(\tilde{\mathbf{x}})}$, we refer to this strategy as latent masking (LM) and the one without any side information as proposed in (Madry et al., 2018) as standard adversarial training (SAT).

We take $k = 50$ in Eq. (3) for estimating $\mathbf{m}$. To evaluate the robustness we employ the PGD attack with 20 steps, denoted as $\mathrm{PGD}_{20}$, and the results are shown in Table 1. Note that, for inference we apply the same mask $\mathbf{m}$ to both

---

[1] In general, this approach is not limited to the penultimate layer.

*Table 1.* Comparison of LM and SAT on CIFAR-10. "Last" and "Best" refer to test accuracy at the end of training, and end of epoch that gives the highest accuracy w.r.t. validation dataset respectively.

| Method | Robust (Last) | Robust (Best) | Clean |
|--------|---------------|---------------|-------|
| SAT | 47.33 | 49.3 | 84.68 |
| LM | 81.63 | 82.8 | 83.18 |

*Table 2.* Test accuracy results for clean and adversarial samples

| Train | Test | Robust | Clean |
|-------|------|--------|-------|
| SAT | $PGD_{20}$ | 47.33 | 84.68 |
| LM | $PGD_{20}$ | 44.66 | 80.31 |
| SAT | $PGD_{20}$ with LM | 39.3 | 84.54 |

the clean and adversarial samples. We observe that when the side information $\mathbf{m}$ is used both during AT and the inference phase, robust accuracy increases by $33 - 34\%$ compared to SAT. We also observe that clean and robust accuracies match at around $83 - 84\%$. This observation empirically supports our claim that using the intrinsic information $\mathbf{m}$ on the consistency of the activation values before the classification step can help to match the clean and robust accuracies.

Next, we perform a complementary experiment to understand the impact of LM on the training and inference phases separately, and the result is shown in Table 2. When LM is employed during training but not at inference, we observe a drop in both clean and robust accuracies compared to when LM is employed for both (cf. Table 1). Hence, using LM during training alone is not sufficient for robustness. However, when we train the network with SAT and apply LM only during inference, we observe that the clean accuracy does not change significantly, while the robust accuracy drops by around $8\%$ compared to SAT. This indicates that with SAT certain latent feature values are useful for the adversarial samples, but may act as noise for clean data. Hence, when LM is not used during training, the network tries to memorize the non-consistent latent features. On the other hand, using LM during training enforces the network to focus on the latent features that are correlated between the clean and adversarial data.

## 4. Side Information with Self-Supervision

We recall that, to obtain LM, we measure the consistency at the penultimate layer $\mathbf{z}^{(\tilde{\mathbf{x}}|\boldsymbol{\theta})}$ by employing an oracle that can access the clean sample $\mathbf{z}^{(\mathbf{x}|\boldsymbol{\theta})}$ as a reference point. Here we try to address the question whether we can generate a "good" reference sample $\hat{\mathbf{x}}$ from the observed sample $\tilde{\mathbf{x}}$ to obtain the mask $\mathbf{m}$ without an oracle, i.e.,

$$\mathbf{m} = S_{topk}(|\mathbf{z}^{(\hat{\mathbf{x}}|\boldsymbol{\theta})} - \mathbf{z}^{(\tilde{\mathbf{x}}|\boldsymbol{\theta})}|). \tag{4}$$

This leads to two main challenges; defining a good reference sample $\hat{\mathbf{x}}$, and generating it from the observed sample $\tilde{\mathbf{x}}$. For this, let $I_c$ and $I_{nc}$ represent the set of indices for consistent and non-consistent activation values, respectively, at the

penultimate layer of a given clean sample $\mathbf{x}$. An activation value $i$ is considered consistent, i.e., $i \in I_c$, if

$$|\mathbf{z}_i^{(\hat{\mathbf{x}}|\boldsymbol{\theta})} - \mathbf{z}_i^{(\tilde{\mathbf{x}}|\boldsymbol{\theta})}| \leq \beta \tag{5}$$

for any pair $\hat{\mathbf{x}}, \tilde{\mathbf{x}} \in \mathcal{B}_\epsilon(\mathbf{x})$ for some small $\beta$ value. Further, we assume that we do not know $I_c$ but only its cardinality, which is $d - k$. Under these assumptions, if there exists a $\tilde{\mathbf{x}}_s \in \mathcal{B}_\epsilon(\mathbf{x})$ such that

$$|\mathbf{z}^{(\tilde{\mathbf{x}}_s|\boldsymbol{\theta})}|_0 = d - k, \tag{6}$$

then one can directly use $\mathbf{z}^{(\tilde{\mathbf{x}}_s|\boldsymbol{\theta})}$ instead of masking the latent values in $\mathbf{z}^{(\tilde{\mathbf{x}}|\boldsymbol{\theta})}$. Even though such an $\tilde{\mathbf{x}}_s$ may not exist, it may still be useful to search for an $\tilde{\mathbf{x}}_s$ with a sparse representation at the penultimate layer $\mathbf{z}^{(\tilde{\mathbf{x}}_s|\boldsymbol{\theta})}$ to use as the reference sample, since adversarial samples increase the non-consistent activation values. Now, to search for $\tilde{\mathbf{x}}_s$, we define a loss function $\mathcal{L}_s(\tilde{\mathbf{x}}, \boldsymbol{\theta}) = |\mathbf{z}^{(\tilde{\mathbf{x}}|\boldsymbol{\theta})}|_1$ and consider a gradient based approach to update the observed sample iteratively, i.e., starting from $\hat{\mathbf{x}}_0 = \tilde{\mathbf{x}}$,

$$\hat{\mathbf{x}}_{t+1} = \Pi_{\mathcal{B}_\epsilon(\tilde{\mathbf{x}})}(\hat{\mathbf{x}}_{t+1} - \alpha \, \mathrm{sgn}(\nabla_{\mathbf{x}} \mathcal{L}_s(\hat{\mathbf{x}}_t, \boldsymbol{\theta}))), \tag{7}$$

where $\Pi_{\mathcal{B}_\epsilon(\tilde{\mathbf{x}})}$ is the projection operator, $\mathrm{sgn}$ denotes the sign operator, and $\alpha$ is the step size.

Hence, $\hat{\mathbf{x}}$ can be generated from $\tilde{\mathbf{x}}$ at inference time by using (7), and we call this procedure as compressive counter-adversarial attack (CCA). By employing CCA, mask $\mathbf{m}$ can be obtained by using Eq. 4, and this overall strategy is denoted as LM with CCA (LM-CCA), where during training clean samples are used as reference samples for LM, and during inference a reference sample is obtained by self-supervision only using the observed sample.

LM-CCA utilizes a fixed masking parameter $k$ for all the samples; however, the number of non-consistent features may depend both on the sample itself and the adversarial attack. To overcome this issue, we propose to use the sample obtained by CCA, i.e., $\hat{\mathbf{x}}$, directly for inference. We show that this strategy, which we refer to as latent compression with CCA (LC-CCA), can also help to increase the robust accuracy compared to SAT. Note that, in both LM-CCA and LC-CCA, the sample $\hat{\mathbf{x}}$ obtained by CCA may not be inside the ball $\mathcal{B}_\epsilon(\mathbf{x})$, which may limit its effectiveness. We investigate this limitation further in Appendix B using the oracle, where we perform additional experiments such that the sample obtained by CCA is always projected to $\mathcal{B}_\epsilon(\mathbf{x})$. Appendix C further extends this to practical scenarios and we observe significant improvements in both the clean and robust accuracies, which highlights the advantage of using compressed representation at the penultimate layer.

Note that, in general, cross-entropy loss $\mathcal{L}_{CE}$ is used for training DNNs, however it makes the classifier over-confident on the target probabilities (Hein et al., 2019; Müller et al., 2019; Szegedy et al., 2016). This over-confidence is particularly an issue for LM since it corresponds to feature selection and the selected features may

*Table 3.* Accuracy comparison for CIFAR-10 dataset. We use the claimed results for CIFS in (Yan et al., 2021)*

| Method | $\gamma$ | Robust (%) | Clean (%) |
|---|---|---|---|
| SAT | 0 | 47.33 | 84.68 |
| CIFS* | - | 51.23 | 83.86 |
| CIFS (FAT)* | - | 51.68 | 86.35 |
| LM-CCA | 0.1 | 63.4 (+16.07) | 82.15 (-2.53 ) |
| LC-CCA | 0.2 | 54.63 (+7.3) | 82 (-2.68) |

change for the samples belonging to same class. This happens because when we use $\mathcal{L}_{CE}$ in LM we enforce the classifier to focus on particular features for each sample. To mitigate this over-confidence we employ *label smoothing*, where we use the target probabilities $\tilde{\mathbf{y}}$, i.e., $\tilde{\mathbf{y}} = (1 - \gamma)\mathbf{y} + \gamma(\mathbf{1}/c)$, where $c$ is the number of classes, $\mathbf{y}$ is a one hot vector of true target class, and $\gamma$ is the label smoothing parameter.

## 5. Experimental Evaluation

We train the ResNet-18 model with different configurations of the LC-CCA and LM-CCA on CIFAR-10 and CIFAR-100 datasets (Krizhevsky, 2009), and similarly to (Rice et al., 2020) we reserve 1000 images from the training set for validation; see Appendix A for full details. We use $\mathrm{PGD}_{10}$ attack during training and $\mathrm{PGD}_{20}$ at inference. We generate counter adversarial samples using Eq. (7).

Table 3 and Table 4 show the robust and clean accuracies for CIFAR-10 and CIFAR-100 datasets, respectively, for the last training epoch. We observe that both of the proposed LM-CCA and LC-CCA schemes can achieve significant improvements of $16\%$ and $7.3\%$, respectively, in robust accuracy compared to SAT on CIFAR-10 dataset. However, we also observe some reduction in the clean accuracy, e.g., $2.5\%$ and $2.7\%$ drop for LM-CCA and LC-CCA, respectively. Furthermore, we observe that the label smoothing parameter $\gamma$ significantly impacts both the clean and robust accuracies. Although finding an optimal $\gamma$ is out of the scope of this work, from the results we consider $\gamma = 0.1$ and $\gamma = 0.2$ as reasonable choices for LM-CCA and LC-CCA, respectively. Additional results with different $\gamma$ values can be found in Appendix D. We also compare our results with a state-of-the-art scheme for suppressing activation values (Yan et al., 2021), denoted by CIFS. We observe that the proposed schemes surpass CIFS in robustness evaluation. We also provide additional results for our proposed approaches using early stopping on the validation set in Appendix D.

Figure 1 shows the convergence behaviour of the LM-CCA and LC-CCA schemes, respectively, on the CIFAR-10 dataset. We observe that LM directly on the activation values (LM-CCA) can increase the instability of training compared to a natural compression strategy in LC-CCA, and we leave further investigation of this as future work.

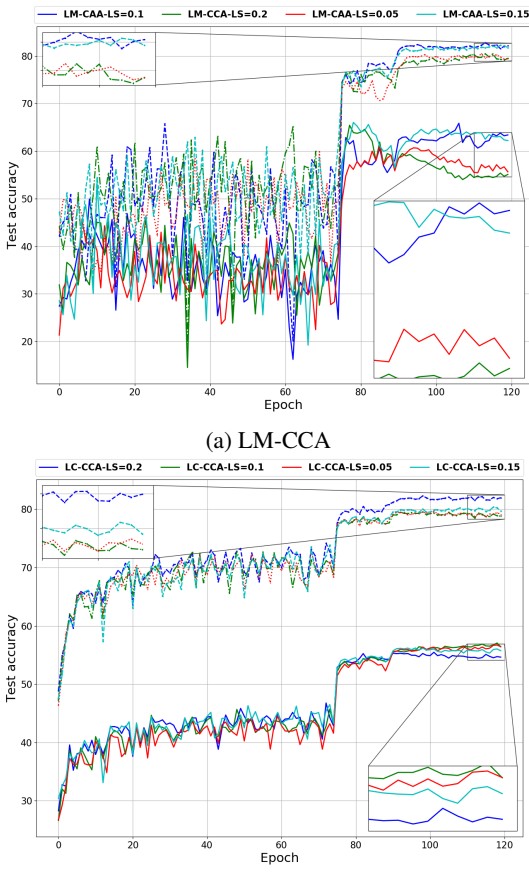

(a) LM-CCA

(b) LC-CCA

*Figure 1.* CIFAR-10 training with different $\gamma$ values for ResNet-18. Test accuracies against $\mathrm{PGD}_{20}$ attack (solid lines) and for clean data (dashed lines) are plotted.

*Table 4.* Accuracy comparison on the CIFAR-100 dataset.

| Method | $\gamma$ | Robust (%) | Clean (%) |
|---|---|---|---|
| SAT | 0 | 23.87 | 58.97 |
| LM-CCA | 0.1 | 28.59 (+4.72) | 57.99 (-0.98) |
| LC-CCA | 0.2 | 29.3 (+5.43) | 57.58 (-1.39) |

## 6. Conclusion

In this work, we showed how a *consistency*-based LM strategy, which corresponds to feature selection, can significantly increase the robust accuracy, and can even match the clean and robust accuracies under supervision of an oracle. *Robust feature selection* is not a new concept in the literature, but the novel aspect of our work is the alternative method devised to identify the robust features by analyzing the consistency at penultimate layer with respect to the perturbations to the input sample. Furthermore, we also introduced the concept of compressive counter-adversarial attack that suppresses the activation values in a self-supervised manner during the inference phase to verify the consistency of the activation values.

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

## A. Experimental Setup

For all experiments, we use ResNet-18 models (He et al., 2016) which are trained using SGD with momentum value of 0.9, weight decay of $5 \times 10^{-4}$ and an initial learning rate of 0.1, which is divided by 10 at the $75^{th}$ and $90^{th}$ epochs with a total training of 120 epochs. Clean images are normalized between values of 0 and 1 and augmented with horizontal flip and random crop, and for label smoothing we consider smoothing parameter values $\gamma \in \{0, 0.05, 0.1, 0.15, 0.2\}$. We set the $\epsilon = \frac{8}{255}$ in $L_\infty$ norm (maximum perturbation) and step size to $\frac{2}{255}$ for PGD attack for all the experiments. For counter adversarial attack, we consider $\epsilon$ and step size same as PGD attack and number of steps are 10 for all experiments.

## B. Additional Results with Oracle

In this section we provide additional results for LM with oracle. We follow the same experimental setup, but now we try to observe the impact of the parameters $k$ and $\gamma$ and discover the limits on the performance of the latent masking strategy. We consider the parameters $k = 50, 75, 100$ and $\gamma = 0, 0.05, 0.1, 0.15$ and the corresponding results are shown in Table 5 for CIFAR-10 dataset. We observe that when $k = 50$ and $\gamma = 0.1$, both clean and robust accuracy can be as high as $89\%$. Similarly, Table 6 for CIFAR-100 dataset shows a high robust accuracy and clean accuracy for $\gamma = 0$ and $k = 100$. These results clearly highlights why feature selection is a promising direction for designing defence strategies against adversarial attacks.

We also argue that the key limitation of the latent compression strategy with CCA is that the image obtained with the counter adversarial attack may end up outside the ball $\mathcal{B}_\epsilon(\mathbf{x})$ and this may limit the prediction accuracy. To verify that we

*Table 5.* Accuracy comparison for CIFAR-10 dataset.

| Method | $\gamma$ | $k$ | Robust | Clean |
|---|---|---|---|---|
| LM | 0 | 50 | 81.63 | 83.18 |
| LM | 0 | 75 | 84.7 | 86.06 |
| LM | 0 | 100 | 81.17 | 84.15 |
| LM | 0.05 | 50 | 83.35 | 82.66 |
| LM | 0.05 | 75 | 78.1 | 80 |
| LM | 0.05 | 100 | 83.83 | 84.06 |
| LM | 0.1 | 50 | **89.02** | **89.45** |
| LM | 0.1 | 75 | 88.76 | 86.03 |
| LM | 0.1 | 100 | 86.91 | 86.27 |
| LM | 0.15 | 50 | 83.17 | 80.81 |
| LM | 0.15 | 75 | 82.73 | 83.99 |
| LM | 0.15 | 100 | 87.31 | 86.04 |

*Table 6.* Accuracy comparison for CIFAR-100 dataset.

| Method | $\gamma$ | $k$ | Robust | Clean |
|---|---|---|---|---|
| LM | 0 | 50 | 49.94 | 66.27 |
| LM | 0 | 75 | 54.8 | **69.9** |
| LM | 0 | 100 | **60.55** | 66.91 |
| LM | 0.05 | 50 | 40.34 | 59.98 |
| LM | 0.05 | 75 | 49.04 | 64.36 |
| LM | 0.05 | 100 | 48.37 | 60.65 |
| LM | 0.1 | 50 | 46.38 | 65.87 |
| LM | 0.1 | 75 | 59.11 | 69.47 |
| LM | 0.1 | 100 | 41.74 | 58.19 |
| LM | 0.15 | 50 | 46.57 | 65.22 |
| LM | 0.15 | 75 | 55.72 | 68.92 |
| LM | 0.15 | 100 | 56.14 | 66.67 |

*Table 7.* Accuracy comparison for CIFAR-10 dataset.

| Method | $\gamma$ | Robust | Clean |
|---|---|---|---|
| LC | 0 | **82** | 85.11 |
| LC | 0.05 | 78.7 | 89.78 |
| LC | 0.1 | 79.3 | 90.48 |
| LC | 0.15 | 74.42 | 90.6 |
| LC | 0.2 | 68.79 | **90.66** |

*Table 8.* Accuracy comparison for CIFAR-100 dataset.

| Method | $\gamma$ | Robust | Clean |
|---|---|---|---|
| LC | 0 | **58.93** | 64.17 |
| LC | 0.05 | 58.64 | 65.57 |
| LC | 0.1 | 58.12 | 65.54 |
| LC | 0.15 | 56.97 | 65.86 |
| LC | 0.2 | 56.15 | **66.82** |

of the LC-CCA scheme such that during the training and inference phases we use a slightly different implementations of LC-CCA. More precisely, we only change it during training where we already employ the oracle that utilizes the information from clean sample. Recall that the counter-adversarial attack searches for a sample $\hat{\mathbf{x}}$ with a sparse representation based on the observed sample $\tilde{\mathbf{x}}$ iteratively in the following manner

$$\hat{\mathbf{x}}_{t+1} = \Pi_{\mathcal{B}_\epsilon(\tilde{\mathbf{x}})}(\hat{\mathbf{x}}_t - \alpha \operatorname{sgn}(\nabla_{\mathbf{x}} \mathcal{L}_s(\hat{\mathbf{x}}_t, \boldsymbol{\theta}))) \qquad (8)$$

starting from $\hat{\mathbf{x}}_0 = \tilde{\mathbf{x}}$. We remark that since $\hat{\mathbf{x}} \in \mathcal{B}_\epsilon(\tilde{\mathbf{x}})$ and $\tilde{\mathbf{x}} \in \mathcal{B}_\epsilon(\mathbf{x})$, $||\mathbf{x} - \hat{\mathbf{x}}||_\infty$ might be as high as $2\epsilon$. Thus, while suppressing the false activation values CCA may also damage useful features. To overcome this issue, we replace the projection $\Pi_{\mathcal{B}_\epsilon(\tilde{\mathbf{x}})}$ used in Eq. (8) with $\Pi_{\mathcal{B}_\epsilon(\mathbf{x})}$ during the training phase. This strategy is particularly helpful for increasing the clean accuracy as shown in Table 9 for CIFAR-10 dataset. We denote this particular strategy with LC-CCA$^\star$. The results show that by employing CCA supervised by oracle during training we can improve both the clean and robust accuracy such that LC-CCA$^\star$ achieves 55.43% robust and 89.85% clean accuracy simultaneously. We also observe that, when the LC-CCA$^\star$ is employed it is more beneficial to use smaller label smoothing parameter $\gamma$. Based on this observation, we further consider a strategy where we use a projection with extra margin parameter $\delta$ during the training i.e., we use the projection $\Pi_{\mathcal{B}_{\epsilon+\delta}(\mathbf{x})}$ in Eq. 8. We refer to this approach as LC-CCA$_\delta^\star$. We also perform experiments with LC-CCA$_\delta^\star$ by taking $\delta = \frac{2}{255}$ and observe that robust accuracy increases around 4% and 12% compared to the LC-CCA$^\star$ and SAT respectively, while losing only 1% in clean accuracy compared to LC-CCA$^\star$ and still 4% higher compared to SAT.

can further improve the test accuracy if $\hat{\mathbf{x}}$ i.e., the sample obtained by CCA can be projected back to ball $\mathcal{B}_\epsilon(\mathbf{x})$, we consider an experiment setup with oracle $o$ such that the oracle $o$ projects $\hat{\mathbf{x}}$ obtained by CCA into $\mathcal{B}_\epsilon(\mathbf{x})$, and corresponding results for CIFAR-10 and CIFAR-100 datasets are shown in Table 7 and Table 8, respectively. It can be seen that as long as one can guarantee that the samples obtained through CCA stays within ball $\mathcal{B}_\epsilon(\mathbf{x})$, then the compression strategy can efficiently cancel the false activation of the adversarial sample as we can achieve up to 80% robust and 90% clean accuracy simultaneously for CIFAR-10 dataset. We obtain similar results for CIFAR-100 dataset and these results indicate that the main limitation of the proposed LC-CCA scheme is ending up with a sample $\hat{\mathbf{x}}$ that is outside the ball $\mathcal{B}_\epsilon(\mathbf{x})$.

## C. Extensions of LC-CCA

Based on superior results that we observed with LC-CCA under the oracle assumption, we propose a further extension

*Table 9.* Accuracy comparison for CIFAR-10 dataset.

| Method | $\delta$ | $\gamma$ | Robust | Clean |
|---|---|---|---|---|
| SAT | - | 0 | 47.33 | 84.68 |
| LC-CCA$^\star$ | - | 0.05 | 55.43 | 89.85 |
| LC-CCA$^\star$ | - | 0.1 | 49 | 90.56 |
| LC-CCA$^\star_\delta$ | 2/255 | 0.05 | 59.15 | 88.51 |
| LC-CCA$^\star_\delta$ | 2/255 | 0.1 | 57.58 | 88.54 |
| LC-CCA$^\star_\delta$ | 1/255 | 0 | 54.49 | 85.97 |
| LC-CCA$^\star_\delta$ | 1/255 | 0.05 | 59.87 | 88.78 |
| LC-CCA$^\star_\delta$ | 1/255 | 0.1 | 55.02 | 90.01 |
| LC-CCA | - | 0.05 | 55.64 | 79.47 |
| LC-CCA | - | 0.1 | 56.86 | 79.12 |

## D. Additional Results For CIFAR-10 and CIFAR-100 Datasets

In this section, we present additional results for different values of label smoothing parameter $\gamma$ for both LC-CCA and LM-CCA schemes in Table 10 for CIFAR-10, and in Table 11 for CIFAR-100, respectively. We also show the best robust and clean accuracy results for the CIFAR-10 in Table 12 and CIFAR-100 in Table 13. Similarly to (Rice et al., 2020), we measure best accuracy according to the validation set. From Table 12, both LM-CCA and LC-CCA frameworks' results show consistent trade off between robust and clean accuracy which is also inline with the Table 3.

*Table 10.* Accuracy comparison for CIFAR-10 dataset.

| Scenario | $\gamma$ | Robust | Clean |
|---|---|---|---|
| SAT | 0 | 47.33 | 84.68 |
| LM-CCA | 0.05 | 55.64 (+8.31) | 79.47 (-5.21 ) |
| LM-CCA | 0.1 | 63.4 (+16.07) | 82.15 (-2.53 ) |
| LM-CCA | 0.15 | 62.21 (+14.88) | 81.75 (-2.93 ) |
| LM-CCA | 0.2 | 55.1 (+7.77) | 79.49 (-5.19 ) |
| LC-CCA | 0.05 | 55.8 (+8.47) | 80.21 (-4.47) |
| LC-CCA | 0.1 | 56.86 (+9.53) | 79.12 (-5.56) |
| LC-CCA | 0.15 | 55.53 (+8.2) | 79.68 (-5) |
| LC-CCA | 0.2 | 54.63 (+7.3) | 82 (-2.68) |

We also plot the convergence behavior of the proposed LM-CCA and LC-CCA strategies for the CIFAR-100 dataset to highlight trade-off for different configurations in Figure 2 . One can easily observe that convergence of the LM-CCA strategy exhibits certain instability, while LC-CCA exhibits comparatively smooth convergence behavior. However, LM-CCA has a better classification accuracy since LC-CCA may eliminate some useful features as well. We conjecture that a hybrid approach that combines both strategies such that instead of masking certain values completely, we can instead use their compressed values (to prevent instability that we observe in case of LM-CCA) will work better and

*Table 11.* Accuracy comparison for CIFAR-100 dataset.

| Scenario | $\gamma$ | Robust | Clean |
|---|---|---|---|
| SAT | 0 | 23.87 | 58.97 |
| LM-CCA | 0.05 | 28.48 (+4.61) | 55.56 (-3.41) |
| LM-CCA | 0.1 | 28.59 (+4.72) | 57.99 (-0.98) |
| LM-CCA | 0.15 | 27.7 (+3.83) | 57.97 (-1) |
| LM-CCA | 0.2 | 30.49 (+6.62) | 52.56 (-6.41) |
| LC-CCA | 0.05 | 27 (+3.13) | 56.27 (-2.7) |
| LC-CCA | 0.1 | 28.36 (+4.49) | 57.64 (-1.33) |
| LC-CCA | 0.15 | 28.86 (+4.99) | 57.51 (-1.46) |
| LC-CCA | 0.2 | 29.3 (+5.43) | 57.58 (-1.39) |

*Table 12.* Accuracy comparison for CIFAR-10 dataset.

| Scenario | $\gamma$ | Robust | Clean |
|---|---|---|---|
| SAT | 0 | 49.3 | 84.91 |
| LM-CCA | 0.05 | 60.62 (+11.32) | 79.79 (-5.12) |
| LM-CCA | 0.1 | 67.1 (+17.8) | 81.67 (-3.24) |
| LM-CCA | 0.15 | 65.25 (+15.95) | 75.33 (-9.58 ) |
| LM-CCA | 0.2 | 64.58 (+15.28) | 74.57 (-10.34) |
| LC-CCA | 0.05 | 56.22 (+6.92) | 78.97 (-5.94) |
| LC-CCA | 0.1 | 56.86 (+7.56) | 79.12 (-5.79 ) |
| LC-CCA | 0.15 | 55.8 (+6.5) | 80.21 (-4.7) |
| LC-CCA | 0.2 | 55.1 (+5.8) | 82 (-2.91) |

we leave it as a research direction for future work.

We further demonstrate the robustness of the proposed LC-CCA and LM-CCA by comparing against the $PGD_{20}$ and $PGD_{40}$ attacks for both CIFAR-10 and CIFAR-100 datasets in Table 14 and in Table 15 respectively. We can see that accuracy against both $PGD_{20}$ and $PGD_{40}$ attacks is almost identical for the proposed frameworks.

## E. Visualization of the Compressive Counter-Adversarial Attack

To illustrate the impact of the compressive counter-adversarial attack, we plot the clean and adversarial images and compare them with image that is obtained by the compressive counter-adversarial attack. Furthermore, we do the same comparison on the distribution of the activation values at the penultimate layer. For this, we consider four images from the test set that are misclassified under adversarial perturbation and classified correctly when CCA is employed. This images along-with the activation values at the penultimate layer are shown in Figures 3, 4, 5, 6.

Based on Figures 4 and 5, subjectively, it is not possible to argue visual improvement with CCA. However, we observe an interesting behavior on the distribution of the activation values on the penultimate layer, that is CCA makes the distribution of the activation values closer to the those observed

*Table 13.* Accuracy comparison for CIFAR-100 dataset.

| Scenario | $\gamma$ | Robust | Clean |
|---|---|---|---|
| SAT | 0 | 28.61 | 55.77 |
| LM-CCA | 0.05 | 28.49 (-0.12) | 55.12 (-0.65) |
| LM-CCA | 0.1 | 30.96 (+2.35) | 56.54 (+0.77) |
| LM-CCA | 0.15 | 33.91 (+5.3) | 49.39 (-6.38) |
| LM-CCA | 0.2 | 31.64 (+3.03) | 45.71 (-10.06) |
| LC-CCA | 0.05 | 28.44 (-0.17) | 56.79 (+1.02) |
| LC-CCA | 0.1 | 29.41 (+0.8) | 53.05 (-2.72) |
| LC-CCA | 0.15 | 30.36 (+1.75) | 55.2 (-0.57) |
| LC-CCA | 0.2 | 31.1 (+2.5) | 55.11 (-0.66) |

*Table 14.* Accuracy comparison for CIFAR-10 dataset.

| Scenario | $\gamma$ | $PGD_{20}$ | $PGD_{40}$ |
|---|---|---|---|
| SAT | 0 | 47.33 | 46.96 |
| LM-CAA | 0.1 | **63.4** | **62.87** |
| LM-CAA | 0.15 | 62.21 | 61.7 |
| LC-CAA | 0.1 | 56.86 | 56.72 |
| LC-CAA | 0.15 | 55.53 | 55.51 |

*Table 15.* Accuracy comparison for CIFAR-100 dataset.

| Scenario | $\gamma$ | $PGD_{20}$ | $PGD_{40}$ |
|---|---|---|---|
| SAT | 0 | 23.87 | 23.66 |
| LM-CCA | 0.1 | 28.59 | 28.6 |
| LM-CCA | 0.15 | 27.7 | 27.64 |
| LC-CCA | 0.1 | 28.36 | 28.39 |
| LC-CCA | 0.15 | **28.86** | **28.81** |

the activation values on the penultimate layer by suppressing false activation values.

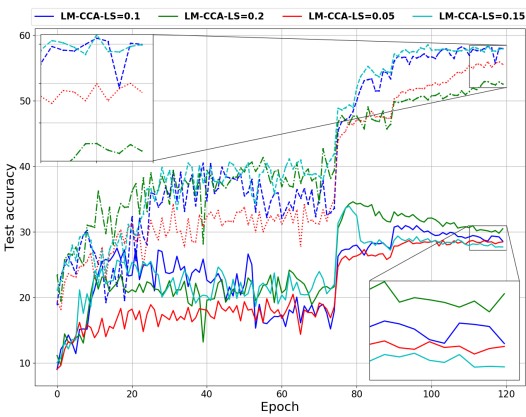

(a) LM-CCA

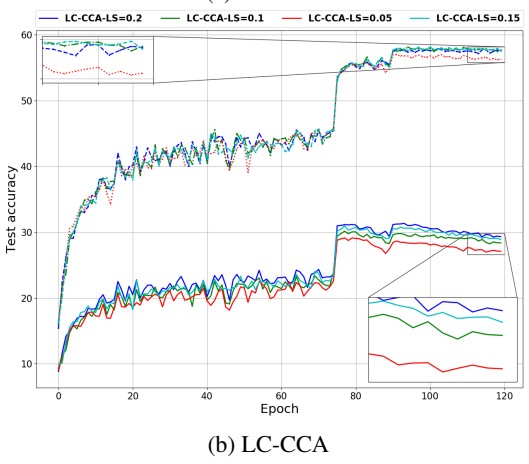

(b) LC-CCA

*Figure 2.* CIFAR-100 training with different $\gamma$ for ResNet-18. Test accuracies against $PGD_{20}$ attack (solid lines) and for clean data (dashed lines) are plotted.

with clean image. To show this correlation quantitatively, we measure the cosine similarity between the activation values of clean and adversarial image as well as the clean image and the image obtained after CCA. We observe that the image with CCA exhibits higher cosine similarity with the clean image. Hence, the compression strategy on the penultimate layer not necessarily work as a purification strategy on the image, but as a regularizer for the distribution of

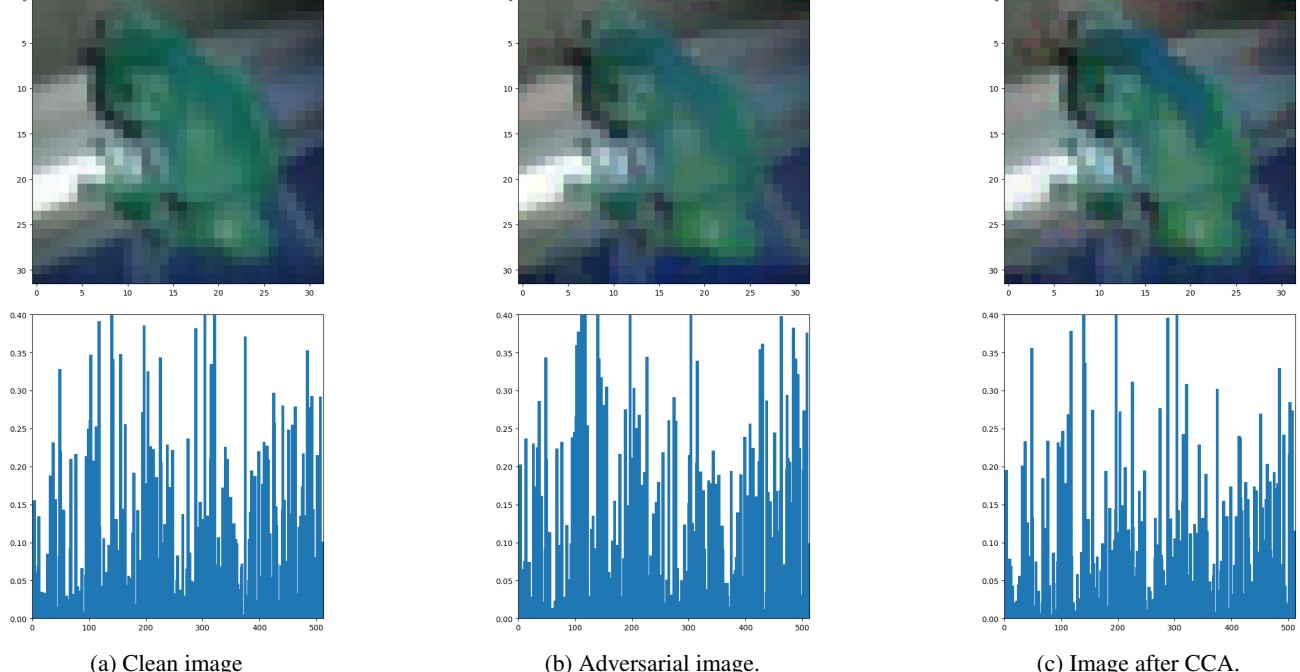

(a) Clean image                    (b) Adversarial image.                    (c) Image after CCA.

*Figure 3.* Comparison between the frog images with their corresponding latent values. Comparing with clean image latent representation, cosine similarity for adversarial latent is 0.84 while counter adversarial one is 0.93. Model's prediction for the counter adversarial image is "frog" but prediction for the adversarial image is "bird".

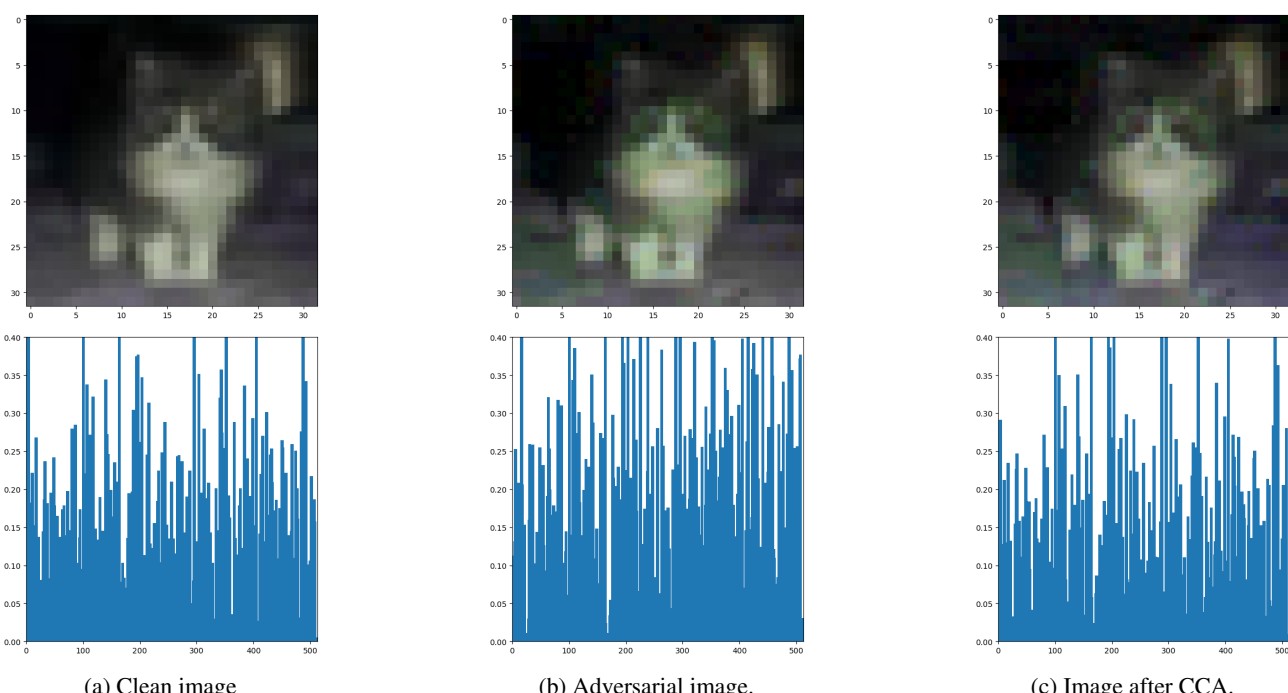

(a) Clean image                    (b) Adversarial image.                    (c) Image after CCA.

*Figure 4.* Comparison between cat images with their corresponding latent values. Comparing to the clean image latent representation, Cosine similarity for adversarial latent is 0.8 while counter adversarial one is 0.95. Model's prediction for the counter adversarial image is "cat" on but prediction for adversarial the image is "frog".

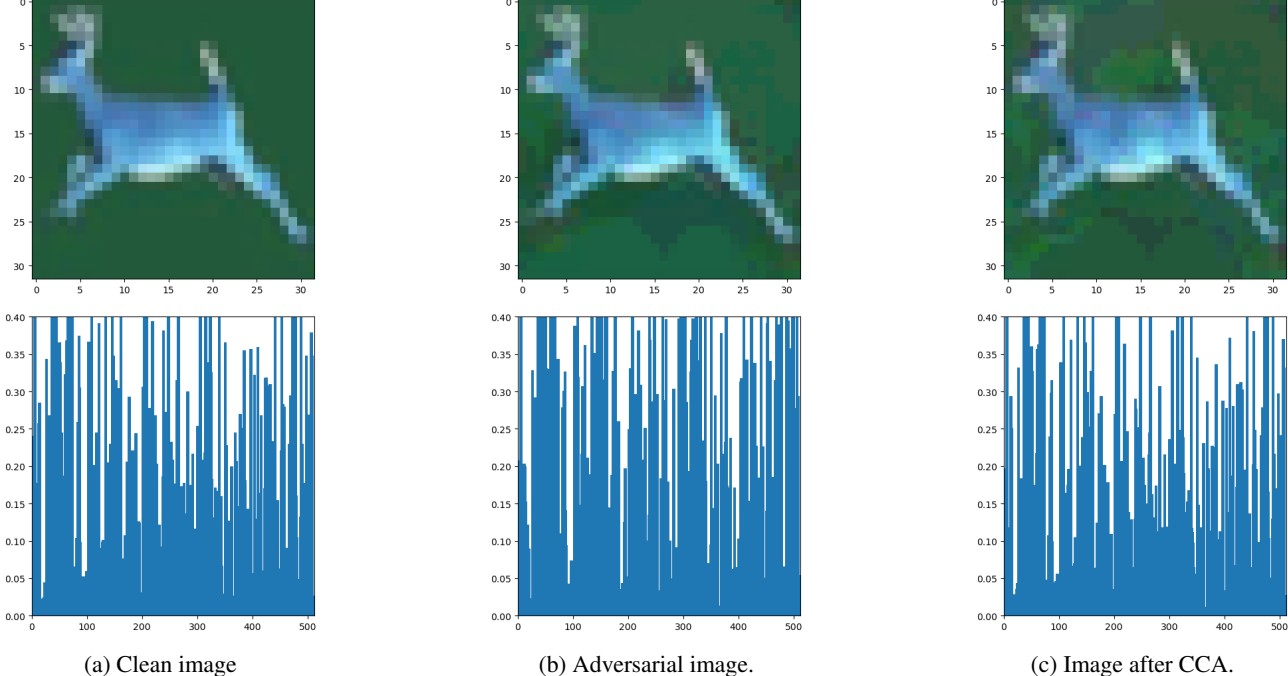

(a) Clean image      (b) Adversarial image.      (c) Image after CCA.

*Figure 5.* Comparison between the deer images with their corresponding latent values. Comparing to the clean image latent representation, Cosine similarity for adversarial latent is 0.88 while counter adversarial one is 0.99. Model's prediction for the counter adversarial image is "deer" on but prediction for the adversarial image is "airplane".

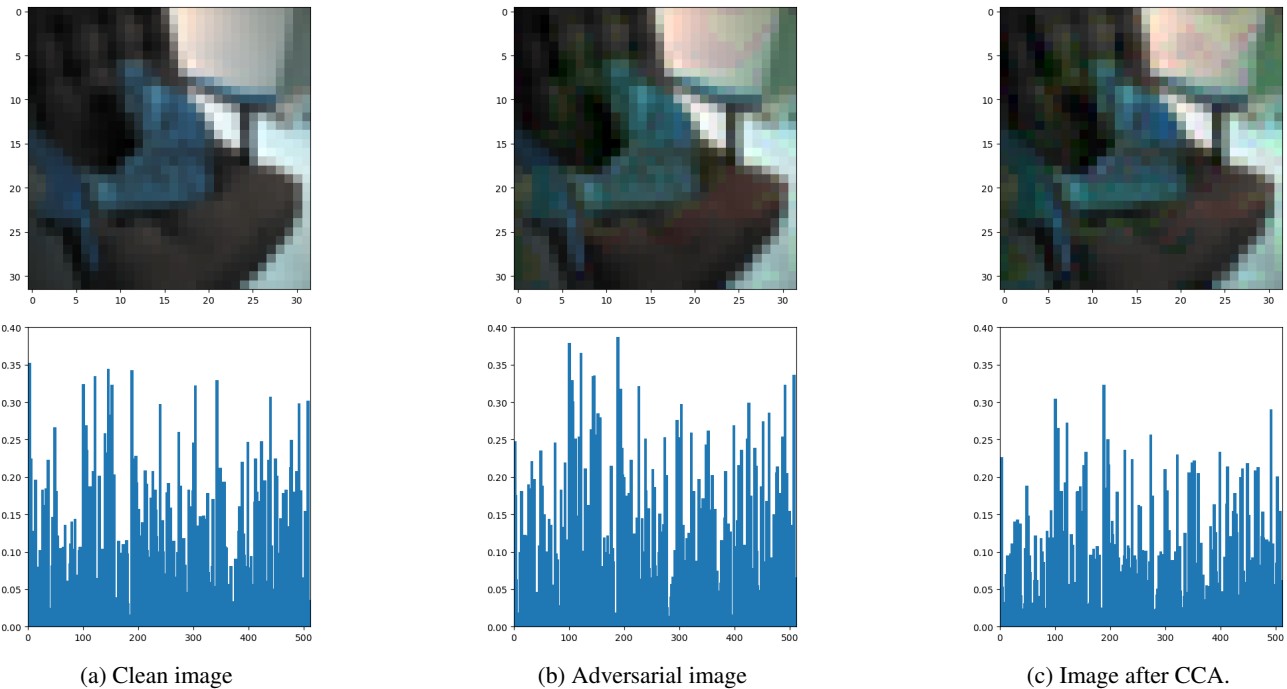

(a) Clean image      (b) Adversarial image      (c) Image after CCA.

*Figure 6.* Comparison between cat images with their corresponding latent values. Compering to the clean image latent values, Cosine similarity for adversarial latent is 0.91 while counter adversarial one is 0.94. Model's prediction for the counter adversarial image is "cat" but prediction for the adversarial image is "bird".