# OpenReview forum: "Less is More: Feature Selection for Adversarial Robustness with Compressive Counter-Adversarial Attacks"
_ICML.cc/2021/Workshop/AML — ICML 2021 Workshop AML Poster_

### Official Review · Reviewer_WjMW · 2021-06-19
**A good idea to do feature selection on penultimate layer to defence. Improve model robustness and clean accuracy.**

**Rating:** Accept
**Confidence:** 4

**Review:**

Pros:
1. The paper is easy to follow
2. The proposed concept is novel
3. The performance is good especially on robustness improvement against the SOTA method.
4. The appendix is very detailed.
Cons:
1. The comparing methods are very limited.

---

### Decision · Program_Chairs · 2021-06-21

**Decision:**

Accept (Poster)

**Comment:**

This paper is well-written and the proposed method is novel. More comparisons may need.